# The Prospective Study of Epigenetic Regulatory Profiles in Sport and Exercise Monitored Through Chromosome Conformation Signatures

**DOI:** 10.3390/genes11080905

**Published:** 2020-08-07

**Authors:** Elliott C. R. Hall, Christopher Murgatroyd, Georgina K. Stebbings, Brian Cunniffe, Lee Harle, Matthew Salter, Aroul Ramadass, Jurjen W. Westra, Ewan Hunter, Alexandre Akoulitchev, Alun G. Williams

**Affiliations:** 1Department of Sport and Exercise Sciences, Manchester Metropolitan University, Manchester M1 5GD, UK; G.Stebbings@mmu.ac.uk (G.K.S.); a.g.williams@mmu.ac.uk (A.G.W.); 2Department of Life Sciences, Manchester Metropolitan University, Manchester M1 5GD, UK; C.Murgatroyd@mmu.ac.uk; 3English Institute of Sport, Nottingham NG12 2LU, UK; briancunniffe@hotmail.com; 4Institute of Sport, Exercise and Health, University College London, London W1T 7HA, UK; 5Holos Life Sciences, Oxford OX1 3HA, UK; lee.harle@holoslifesciences.com; 6Oxford BioDynamics, Oxford OX4 2JZ, UK; matthew.salter@oxfordbiodynamics.com (M.S.); aroul@oxfordbiodynamics.com (A.R.); willem.westra@oxfordbiodynamics.com (J.W.W.); ewan.hunter@oxfordbiodynamics.com (E.H.); alexandre.akoulitchev@oxfordbiodynamics.com (A.A.)

**Keywords:** genetics, epigenetics, exercise, exercise training, chromatin, sports medicine

## Abstract

The integration of genetic and environmental factors that regulate the gene expression patterns associated with exercise adaptation is mediated by epigenetic mechanisms. The organisation of the human genome within three-dimensional space, known as chromosome conformation, has recently been shown as a dynamic epigenetic regulator of gene expression, facilitating the interaction of distal genomic regions due to tight and regulated packaging of chromosomes in the cell nucleus. Technological advances in the study of chromosome conformation mean a new class of biomarker—the chromosome conformation signature (CCS)—can identify chromosomal interactions across several genomic loci as a collective marker of an epigenomic state. Investigative use of CCSs in biological and medical research shows promise in identifying the likelihood that a disease state is present or absent, as well as an ability to prospectively stratify individuals according to their likely response to medical intervention. The association of CCSs with gene expression patterns suggests that there are likely to be CCSs that respond, or regulate the response, to exercise and related stimuli. The present review provides a contextual background to CCS research and a theoretical framework discussing the potential uses of this novel epigenomic biomarker within sport and exercise science and medicine.

## 1. Introduction

Discovering and quantifying the physiological response to external stimuli is a cornerstone of sport and exercise science [1]. Specifically, exercise scientists are interested in the transitory response to exercise and the process by which this translates to long-term adaptation. In recent decades, exercise physiologists have shifted focus toward understanding how changes in whole-body phenotypes are underpinned by adaptations at the cellular and subcellular levels [2]. In the 1970s and 1980s, the introduction of biochemical techniques in sport and exercise science enriched the understanding of cellular function during and after exercise, preceding the recent shift toward molecular techniques and the emergence of molecular exercise physiology [3]. A combination of advancing technology and lower cost now allow the study of whole-body, cellular and molecular responses to physical activity simultaneously. Thus, molecular exercise physiology is credited with complementing and improving ‘classical’ exercise physiology approaches [4] and is likely to accelerate further development in the field of sport and exercise science.

Human phenotypes (traits) are determined by the selective utilisation of an individual’s unique genotype (DNA sequence) upon exposure to environmental stimuli, such as exercise [5]. When exercise stimuli are experimentally controlled, considerable variation exists in the way that individuals respond, suggesting that there is a significant influence of genetic factors [6]. Consequently, sport and exercise genomics has emerged as a sub-discipline of molecular exercise physiology, and aims to understand the organisation and function of the genome in athletic settings [7]. Sports genomics typically explores whether common DNA sequence variants, such as single nucleotide polymorphisms (SNPs) [5], are associated with a phenotype of interest. Indeed, genomic differences exist between individuals that may explain the variance in skeletal muscle fibre type [8], aerobic capacity [6] and muscle strength [9], and there are several specific genotypes associated with the achievement of elite athlete status [7,10]. Genetic approaches such as linkage analysis, candidate gene studies, genome-wide associated studies (GWAS) and whole exome or genome sequencing, while able to test the relationship between fixed genetic variability and quantifiable traits [11], are less able to also account for and understand the role of the environment. Another level of biological regulation, known as epigenetics, is recognised as being essential to the control and programming of gene expression [12]. While heritable genetic factors contribute to approximately 50–60% of exercise-related traits [6,8,13,14,15], with the environment contributing the remainder, epigenetic mechanisms describe the integration of the two and enable the control of gene activity without changes in DNA sequence [12,16,17,18]. As such, the epigenetic regulation of genes in response to external stimuli is an important modulator of physiological adaptation, and epigenetic research is gaining interest in sport and exercise to understand how cells, tissues and organs are able to adapt to exercise, training and related physiological challenges [19].

Growing interest in fundamental genetic and epigenetic mechanisms has driven new directions in sport and exercise science, with technological advances in molecular biology allowing researchers to embrace new techniques and generate new hypotheses in a more applied way. Chromosome conformation is an epigenetic modification that has gained interest in biological and medical research following evidence that some three-dimensional arrangements within the human genome are dynamic in response to external stimuli [20], whilst other conformations can modulate responses to external stimuli without structural rearrangement [21]. However, nothing is known about whether changes in chromosome conformation occur in response to exercise and related stimuli. The aims of this review are to summarise the importance and translation of the human genome to sport and exercise science, and to discuss the potential of a recently developed measure—the chromosome conformation signature (CCS)—to explore the dynamic nature of epigenetics in sport and exercise science.

## 2. Importance of the Genome to Human Biology

The human genome is the complete set of nucleic acid sequences, encoded as deoxyribonucleic acid (DNA), which are around 99.9% identical between individuals [22]. Comprised of around 3 billion base pairs (bp), only around 1–2% of the genome contains sequences for protein-coding genes. These genes contain sequences transcribed to ribonucleic acid (RNA), which is translated to form proteins, which make up the structure of organs and tissues, control chemical reactions, and transport signals between cells [23]. Accordingly, the information stored as DNA is the essential foundation for the development and function of the human body [24].

The remaining 98–99% of non-coding DNA was once considered “junk”, as it had no known functions. However, it is now appreciated that much of this non-coding DNA is crucial to the functioning of cells, especially maintaining the stability of the genome and regulating the activity of genes. Much of this non-coding DNA is transcribed into non-coding RNA with various regulatory activities, whilst there are also DNA sequences that act as regulatory elements, determining when and where genes are turned on and off. Depending on their functions and locations, these include promoters, enhancers, silencers and insulators [25]. Tight regulation and plasticity in the expression of specific genes during and/or following exercise is central to adaptation [26], with the mechanisms that govern the regulation and function of the human genome having particular relevance to sport and exercise science and medicine.

Genomic DNA is packaged by tightly wrapping around histone proteins in a complex known as chromatin [27] to allow the almost 2 m of linear DNA in each human cell to fit into the nucleus. Chromatin has a repeating and functional subunit known as a nucleosome, consisting of eight histone proteins and approximately 146 DNA bp, and regulates the access of transcriptional machinery to DNA that must unravel before transcription [28] (Figure 1). This unravelling, known as chromatin remodeling, is an epigenetic process regulated by cell-specific histone modifications, such as methylation, acetylation and phosphorylation, that mark genes, transcription start sites and stretches of regulatory DNA to control gene expression [27]. Near the start of a gene is the core promoter, serving as a docking site for RNA polymerase II and transcription factors to form the transcription pre-initiation complex. Approximately 250 bp upstream lies the proximal promoter, which is a sequence containing primary regulatory elements where general transcription factors bind. Enhancer regions are often located hundreds or thousands of bp away from transcription start sites, and are short stretches of DNA that can increase the transcription of genes [25]. As these regulatory elements of the DNA sequence are not adjacent to one another, loops in the DNA bring distal enhancer regions closer to the proximal promoter. Functional organisation of the genome is, therefore, not simply linear along chromosomes, as DNA elements can regulate genes located far away on the same (intrachromosomal) or different (interchromosomal) chromosomes, due to the organisation of the genome in three-dimensional nuclear space [29,30] (Figure 2). Importantly, the capacity for distal chromosomal regions to enter close physical proximity and facilitate interaction of DNA elements makes spatial chromatin organisation a key mechanism in regulating gene expression [17,31,32]. Critically, genome packaging and rearrangement is non-random, with spatial chromatin organisation assuming a key functional role in the retrieval and translation of genetic instructions [33]. While the analysis of spatial chromatin organisation has the potential to provide valuable insight into the role of genome packaging on gene expression in various settings, how this affects the expression of genes involved in the response to exercise and related stimuli is unknown.

## 3. Assessment of Chromosome Conformation

Determining how chromosomes are positioned and organized within the nucleus, and how this positioning changes with respect to environmental stimuli, is critical to understanding the clinical relevance of chromatin architecture. Over the years, several methods which vary in their level of molecular insight, resolution, and technical complexity have been developed to study chromosome architecture, with the most well-developed techniques based on chromosome conformation capture (3C) [29,34]. The seminal report on the use of 3C was described in 2002, where researchers used a yeast model system to develop a methodology to detect the frequency of interaction between any two genomic loci [35]. The basis of 3C is to identify and quantify the number of interactions between genomic loci, potentially separated by many thousands of nucleotides in the *linear* genome, within three-dimensional space [36]. In laypersons’ terms, the 3C methodology “freezes” the 3D genome in time and asks “which genomic regions are interacting with each other?” Operationally, the basic 3C method involves fixation of cells/nuclei with the chemical agent formaldehyde to cross-link DNA and protein interactions, which stabilizes interactions between genomic loci. Next, the genome is fragmented using restriction enzymes, which cleaves DNA into fragments at or near specific recognition sites in the genome. Following genomic fragmentation, a proximity-based ligation reaction step under conditions that favour re-ligation between cross-linked interacting fragments that were physically proximal during fixation over re-ligation between fragments that are not cross-linked (genomic regions that were physically distant during fixation). This results in a stable 3C ‘library’ of interactions which can be detected and quantified using several different approaches, primarily PCR-based methods and next generation sequencing. This composition of this 3C library can vary depending on the restriction enzymes used and yields a way to identify and quantify chromosomal interactions as a starting point. In addition, the PCR methods used to detect interactions from a 3C library requires prior knowledge of the targets of interest. Since the initial description of the 3C approach, several methodological variants have been developed to study genome architecture, including circular chromosome conformation capture (4C), chromosome conformation capture carbon copy (5C), combined chromosome conformation capture ChIP cloning (6C), ChIA-PET, Hi-C, and RNA-TRAP that are well summarised elsewhere [29,30,37,38]. Each technique varies by cost, resolution, bandwidth and throughput, and can be used in combination to provide a view of the genome in vivo for research use, though not a clinical quality assay [29]. The resulting view of spatial chromatin organisation is used to determine the expression state of genes regulated by physical contacts in the nuclear space.

Chromosome conformation signatures (CCSs) are collections of multiple DNA contacts associated with specific functional outcomes such as disease states and gene expression patterns [29], and technologies for identifying such signatures have also been developed. Recently, an approach with a focus on clinical and industrial scale screening and applications was developed [39,40,41,42]. Like the original 3C methodology, *EpiSwitch™* detects the absence or presence of interactions between two sites in the genome. However, a key difference in the approach is that unlike traditional 3C, *EpiSwitch* leverages in silico knowledge of interaction sites within the genome to guide the interactions that are evaluated. In essence, the approach asks “which of the key regulatory interactions exist?” in a given sample. Another difference between *EpiSwitch* and conventional 3C analysis is scale. While only single interaction pairs at a time are identified by PCR in 3C, initial *EpiSwitch* screening is done via a comparative genomic hybridization (CGH) array, with the ability to assess the absence or presence of over 1 million regulatory interaction pairs at a time. This initial array-based screening step can then be followed by PCR or sequencing-based detection for interactions of interest.

At present, CCSs are not commonly applied in sport and exercise science and medicine, but may offer a viable method to identify the epigenomic factors underpinning responses and adaptations to exercise and related stimuli. Though the terms chromosome and chromatin are used interchangeably in the related literature, in this review we will refer to chromosome conformation signatures.

## 4. Chromosome Conformation Signatures (CCSs)

Chromosome conformation signatures represent a novel epigenetic biomarker of structural epigenetic changes in genomic architecture, documenting collections of DNA contacts associated with specific physiological outcomes [29]. With multiple genomic loci contributing to phenotypic differences [43], a signature of multiple DNA contacts is likely to provide greater biological insight than a single contact alone. An analogy would be how singular factors underpinning running performance (VO_2max_, lactate threshold, running economy, VO_2_ kinetics) provide better informative value when analysed in combination [44]. Similarly, the polygenic influence on physical performance [45] lead scientists in the field of sports genomics to consider the combined influence of multiple genetic variants [46] as opposed to single genomic loci. That the expression of numerous genes is increased/decreased according to function and drives physiological adaptation [26] leads us to hypothesise that there are detectable conditional CCSs associated with acute and chronic responses to exercise and related stimuli. Specifically, the dynamic nature of chromatin organisation [20] suggests there may be signatures associated with both transient responses and more persistent phenotypic changes, reflecting the underlying epigenetic regulatory landscape. Recent evidence that transcription factor activity, in particular, is affected by promoter region interactions [21,47] demonstrates the permissiveness of CCSs to facilitate acute and persistent physiological alterations. Recent application of CCSs [39,40] has shown that this biomarker modality can be applied to whole blood samples [41] to provide stable, binary readouts between two states (pre-intervention vs. post-intervention, disease vs. non-diseased) based on the presence or absence of a signature [29,39,42]. It is important to note that study of CCSs requires consideration of participants individually, as opposed to their contribution to a collective group mean. Mean values are routinely calculated and reported in sport and exercise research to summarise group data, providing a measure of central tendency [48]. However, extreme values influence group means, particularly in small samples, and are less suited to investigating variability between individuals. The binary nature of CCS readouts offers a different approach, with samples (participants) grouped according to the presence or absence of a specific signature, as opposed to each sample contributing a numerical value on a scale, such as when measuring VO_2max_ or jump height. Hence, CCS technologies identify the flexibility or inflexibility of epigenomic *states*, rather than reporting the *magnitude* of gene expression, across multiple genomic loci. Importantly, the way in which CCSs reorganise in response to stimuli to regulate gene expression appears to be one of the earliest detectable events, preceding other epigenetic modifications, transcription factor binding and transcription [49]. Due to the fact that DNA is spatially organized into 3D structures, and distal genomic regions can be brought into proximity through chromatin folding, it would be expected that such DNA sequences may also exhibit coordinated epigenetic marks, such as histone modifications and DNA methylation. Indeed, a recent study using a variation of CCS, termed Methyl-HiC, revealed coordinated DNA methylation status between distal genomic segments that are in spatial proximity in the nucleus [49]. Such combined approaches would be important to understand how epigenetic marks are dynamically regulated with characteristic patterns in different tissues. The ability to detect these early molecular changes may provide considerable benefit to sport and exercise scientists who seek to understand the initial drivers of adaptation.

## 5. Studies Using CCSs in Biomedical Research

The potential of CCS has emerged in the last decade, with a number of human studies in medicine demonstrating promising results [39,40,41,42,50]. An intriguing advantage is that CCSs can indicate the likelihood that a specific condition or disease is present [39,40], or can be used to identify individuals who are likely to be responsive or non-responsive to medical intervention [41]. Importantly, the diagnostic concepts of sensitivity and specificity are common to each approach, describing the ability of a test to identify those with or without a given disease or trait [42,51]. Although sport and exercise scientists do not typically investigate disease or therapeutic intervention, medical research provides valuable insight into the potential use of CCS in exercise-related settings as part of a move towards more targeted approaches and personalised interventions.

One of the first studies to assess the ability of a collective ‘signature’ of contacts (markers) to detect the presence or absence of a disease was in oncology. After identifying loops associated with specific candidate genes, researchers explored whether a CCS could distinguish between patients with melanoma and non-melanoma skin cancers [40]. Eighty-six genomic loci were identified as potential regions of chromatin investigation and screened for chromatin loops in blood samples from melanoma patients. From this screen, a panel of the 15 most informative and robust melanoma markers was identified. The panel successfully discriminated between melanoma and non-melanoma samples with 81.4% sensitivity and 76.5% specificity, and could successfully identify non-melanoma blood samples from healthy controls with 87% accuracy. Consequently, the binary state of chromosome conformations was proposed as a quick, non-invasive test for melanoma, potentially informing treatment, as well as detecting pre-symptomatic molecular changes that could improve diagnosis and prognosis [40]. More recently, Salter and colleagues identified a blood-based CCS that discriminated between amyotrophic lateral sclerosis (ALS; a form of motor neurone disease) patients and healthy controls [39]. In a multi-step selection process, a microarray-based screen on 13,880 potential conformations across 308 genomic loci specific to the immune-footprint of ALS was used to select the chromosomal interactions that best discriminated ALS from healthy controls. From this initial pool, 153 interactions were translated to polymerase chain reaction (PCR)-based detection and evaluated over multiple rounds of marker selection/reduction on increasing numbers of patient samples, until the top eight interactions were identified and used to create a CCS. The CCS discriminated patients from controls with 83.3% sensitivity and 76.9% specificity and provided proof-of-concept that changes in genomic architecture in blood samples could reliably detect physiological changes associated with a disease. Ongoing investigation into the role of CCSs in ALS is currently being studied in a clinical trial setting (ClinicalTrials.gov, NCT04259255). These studies demonstrate the capacity to determine presence or absence of a specific disease by CCS, which is particularly important given that chromosomal conformation is the smallest unit of genome regulation linked to phenotype [29,43] and that conformational changes are amongst the earliest detectable events [29,49]. The latter is especially pertinent in conditions like ALS, where early symptoms are subtle and may lead to delayed diagnosis [39].

Identifying individuals who are more/less likely to respond to a treatment intervention represents another attractive use of CCSs, where predicting the response to treatment can be highly challenging. A hypothetical example of a process to enable patient selection for drug response is depicted in Figure 3. Recently, Carini and colleagues described a CCS of inadequate response to methotrexate (MTX), a drug used to treat rheumatoid arthritis (RA) [41], in early RA patients. Blood samples taken from early RA patients prior to treatment were used to screen 13,222 chromosomal interactions across 123 genetic loci previously associated with RA to identify a ‘non-response’ signature. Taking a similar stepwise approach with logistical regression to refine the CCS, a CCS consisting of five markers was identified and correctly classified 90% of existing patients as responders or non-responders to MTX, with 86% specificity in a blinded test cohort. As drug responsiveness is a significant predictor of long-term outcome in RA [52], this study demonstrated the value of using a CCS-based biomarker approach to identify patients’ likelihood of response to medical intervention.

## 6. Features of CCSs Applicable to Sport and Exercise Science and Medicine

Biomarkers are characteristic biological properties that can be detected in blood or tissue to indicate a physiological or pathological process [53]. Biomarker analysis is routine in modern sport and exercise medicine research, with many institutions investigating cells, molecules, genes, enzymes and hormones via multiple techniques. The potential to include novel data concerning the earliest detectable events, within the smallest detectable unit of regulated genome [29], make CCSs an attractive tool to complement existing knowledge on the mechanisms governing genome regulation in response to exercise and related stimuli.

A notable advantage of CCSs is the link between epigenetic regulation and phenotype [39]. Whilst GWAS studies highlight genomic loci associated with phenotypes, causality is largely unexplained, and epigenetic variation is not quantified [54]. Epigenome-wide association studies (EWAS) are the epigenomic equivalent and study the proportion of phenotypic variance attributed to epigenomic variation. However, interindividual differences identified by EWAS such as DNA methylation could be either the cause or consequence of a phenotype [54]. While CCSs also have the potential to be the cause or consequence of a phenotype, they integrate both genomic and epigenomic regulation of gene expression at the smallest detectable level [43], allowing the identification of genomic loci associated with, and therefore indicative of, specific physiological states.

Notably, a CCS provides results on a binary basis based on the presence/absence of a specific three-dimensional architecture for a given phenotype, as opposed to quantifying the magnitude of difference between individuals or groups, allowing efficient stratification of samples (Figure 3). Though CCS does not measure gene expression levels that may provide detail of subtle differences between levels of stimuli (e.g., duration and/or intensity of exercise), investigating CCS and gene expression simultaneously can circumvent this limitation to provide greater insight than each alone. Binary outputs are common amongst studies of genetic variants, such as SNPs—the result is binary, according to the presence or absence of the allele. This, however, only applies when investigating a single SNP, which has limited value due to the highly polygenic nature of physiological traits [45]. Furthermore, SNPs are stable, heritable genomic features that may be more or less influential depending on other inherited DNA sequence variations and gene-environment interactions, whereas CCSs can dynamically reorganise in response to major external stimuli to regulate gene expression across several genomic loci. The contribution of several genomic loci to each CCS offers superior investigative value than single variants or conformations, with each conformation identified and refined through rigorous statistical selection, suggesting that sport and exercise science research participants could be stratified effectively by CCS in a number of contexts.

Within sport and exercise science, where efforts are made to predict the magnitude of training response [55,56] or the likelihood of injury [57] using genetic association data, results are mixed, which may be due to heterogeneity in genetic background, among other confounding factors. However, the iterative discovery and refinement stages employed when using CCSs offers a more rigorous approach, with the predictive ability and signature validation maintained in independent cohorts reflecting this [39,40,41]. Furthermore, sample sizes in existing CCS research are relatively modest in comparison to typical genetic studies, where larger sample sizes are recommended to achieve sufficient statistical power [58]. In CCS research, initial screening for significant markers is performed using small subsets of samples, with larger sample cohorts used in subsequent screening stages to achieve the statistical power needed to approximate real-world populations [39]. Modest sample sizes, predictive accuracy and the inclusion of multiple genomic loci make the use of CCSs in exercise-related research attractive.

An important aspect of a useful biomarker is clinical accessibility. The source of the measurement must be easy to obtain non-invasively and provide a useful and biologically meaningful readout. For these reasons, peripheral blood is a common biomarker source in sport and exercise science. Peripheral blood has been used as the primary biofluid source of existing studies on CCS in clinical applications, with the added benefit of providing a systemic readout of physiological changes. CCS readouts also require minimal material, with as little as 50 µL of whole blood [41] routinely used. This makes sample collection itself relatively straightforward, and highly applicable to exercise-related studies where blood samples are collected to study a range of analytes. Relatively few studies have systematically assessed the concordance of changes in genomic architecture between primary tissue and peripheral blood. However, initial studies indicated that shared epigenetic characteristics, including chromatin structures, between the primary tissue and surrogate blood-based readouts exist [40,59,60]. Nevertheless, it is important to consider that changes in blood cell homeostasis should also be considered or controlled for, especially in acute studies. While the exact molecular underpinnings of shared chromatin structures are being actively researched, horizontal RNA transfer—an exosome-mediated resetting of selected targeted cell populations—is proposed as one [61]. Exosome-based transfer of non-coding RNA and metabolic signalling is involved in the epigenetic resetting of secondary cellular targets in distal tissues and in peripheral blood and, crucially, is directly associated with the resetting of specific chromosome conformations in individual effector cells [39]. Appropriate study design is, therefore, imperative to have confidence that any observed changes detected by a CCS (such as pre- and post-intervention) can be attributed to a defined stimulus. Lastly, while the current cost of CCS detection technologies may limit initial uptake to institutions/organisations able to access/direct substantial funding towards such work, technological advances and efficiency gains in scale made by specialist laboratories that perform CCS analysis should reduce costs in time.

## 7. Potential Use of CCSs in Sport and Exercise Science and Medicine

Existing studies have suggested that CCSs are a valid and reliable method to identify epigenomic state and changes associated with specific physiological stimuli. Thus, pre-existing CCSs may influence the nature and extent of acute responses to exercise and/or chronic adaptations to training. Additionally, rearrangement of three-dimensional chromatin structure to regulate gene expression under external perturbations [62] suggests that alterations in CCSs may occur in response to acute exercise or chronic training. Indeed, the chemical, electrical and mechanical signals generated by exercise are converted to molecular events that promote physiological responses, culminating in the activation and/or repression of various signalling pathways regulating gene expression [63]. Furthermore, a growing body of research in human and animal models suggests that epigenetic modifications, including DNA methylation, histone modifications and microRNAs, are bi-directionally associated with responses to aerobic and resistance exercise in numerous tissues including the brain, blood, skeletal muscle, adipose tissue and epithelial buccal cells [64]. However, the regulation of gene expression and epigenetic marks by chromosome conformations in response to exercise remains unknown, presenting a number of scenarios where the exploration of CCSs may be of interest to sport and exercise researchers and, subsequently, practitioners.

### 7.1. CCSs of Response to Single Exercise Bouts

Changes in gene expression occur after single bouts of exercise [65]. When reinforced by repeated exercise stimuli, lasting and cumulative effects on gene expression at the protein level form the basis of phenotypic adaptations that contribute to improved performance [26]. Elevated mRNA [66] and hypomethylation of genes [67] are reported after a single bout of exercise, whilst expression of other genes can be downregulated by exercise [26]. Nevertheless, the acute effect of exercise on chromatin conformation is currently unknown, and the analysis of CCSs may provide valuable insight regarding the regulation of multiple genomic loci in response to exercise.

A protocol involving pre- and post-exercise blood samples, plus a series of follow-up blood samples (for example at 24, 48 and 96 h post exercise) would allow the exploration of CCSs in response to a single exercise bout. The influence of training status [68] and macronutrient availability [69] on mRNA responses to acute exercise should be appropriately controlled, with a non-exercise control group also necessary. The aim of singular-bout studies would be to identify a CCS to distinguish between pre- and post-exercise epigenomic states, which would provide insight into the temporary rearrangement and reorganisation of chromatin conformations following exercise. Identifying such loci would help sport and exercise researchers to understand the earliest stages of physiological adaptation, as well as aiding the development of new targets for gene expression analyses. In such instances, identification of key loci could be combined with mRNA profile analysis to provide a more detailed mechanistic explanation of the adaptive response to training, with genomic interaction data lacking an indication of gene expression magnitude or splice isoform behaviour. To explore how CCSs respond to single bouts, separate endurance and resistance studies are recommended due to the different molecular programmes elicited by each modality [70]. However, the inability to extrapolate observations from single bouts to the phenotypic adaptations to long-term training is a limitation. Consequently, training intervention studies are required to determine how CCSs respond to repeated exercise stimuli and regulate training adaptation.

### 7.2. CCSs and Exercise Training

#### 7.2.1. Adaptations of CCSs to Training

Adaptations of CCSs in response to sustained training may differ from their response to acute exercise. Training studies will, therefore, help to ascertain whether CCSs are associated with changes in exercise-related phenotypes that occur following persistent stimuli, and how CCSs relate to altered gene expression profiles and epigenetic modifications that underpin these changes [71,72]. Exploring whether phenotypic adaptations are associated with the rearrangement of CCSs during training adaptation requires pre- and post-training blood samples, and sequential sampling throughout training could establish the time course of CCS changes during an intervention. Specific criteria to define a training adaptation phenotype may involve a minimum percentage change relative to pre-training. Pre- and post-intervention blood samples from participants exhibiting the training adaptation phenotype can be screened for chromosomal interactions that distinguish between them to identify a CCS. After statistical refinement to detect the most discriminatory interactions, accuracy of the CCS can be tested by randomising pre- and post-training samples into groups and observing the ability of the CCS to correctly discriminate between them. Validation of the signature can be achieved using an independent, blinded cohort of participants and assessing how well a CCS determines whether samples were from trained or untrained participants. Discovering whether specific CCSs adapt to training (or detraining) and are associated with phenotypic changes has the potential to improve current understanding of the mechanisms that underpin physiological adaptation.

#### 7.2.2. Using CCSs to Predict Training Adaptations

Training studies typically quantify the magnitude of adaptations by groups. However, group data can conceal the variability in the way individuals adapt [73], meaning individual data are important when investigating variable training adaptations. To determine whether individual adaptations to training (or detraining) are regulated by CCSs that are present prior to an intervention, variable individual adaptations must be categorised.

The optimal suggested study design for quantifying individual differences in adaptation is a replicate crossover design (control and exercise conditions administered to participants at least twice in randomised order) [74], enabling differentiation between true individual differences and those attributable to measurement error. Replicate crossover studies are rare in sport and exercise [75], undoubtedly due to the challenges of conducting such trials. However, a potential alternative is for participants to repeat a training intervention after a washout period, allowing the determination of true individual differences in adaptation that are difficult to quantify using a single intervention [75] (Figure 4a). These comments regarding study design also apply to the later sections of this manuscript regarding nutritional and other interventions, but will not be mentioned there to avoid repetition. After the interventions, participants can be ranked according to the magnitude of improvement from pre-intervention from the largest to smallest change, allowing subsequent exploration of whether CCSs are involved in regulating the observed adaptation. Participants ranked in the top and bottom 25% could be classified as ‘high’ and ‘low’ adapters, respectively, with pre-intervention samples from each group compared in a ‘discovery’ phase [41] to identify chromosomal interactions associated with high and low adapter phenotypes. The discovered CCS is refined cyclically by increasing the number of pre-training samples from high and low adapters to define the best discriminatory interactions. Next, randomised pre-training samples would allow the testing of the CCS’s sensitivity and specificity to distinguish between high and low adapters, with the signature’s predictive ability validated by prospectively stratifying the pre-training samples of independent participants and observing how individuals adapt to training (Figure 4b).

Once experimentally validated, the use of CCS to make predictions about individualized training adaptations could complement other efforts to identify talent and individualise training programmes. Further directions include investigating whether CCSs regulate (and can predict) general ‘trainability’ regardless of exercise type, or if specific CCSs are associated with whether individuals who adapt well to endurance exercise are less responsive to resistance exercise, and vice versa. The heritable nature of exercise trainability [6,76] and muscle fibre composition [8] suggests that some individuals are inherently suited to specific exercise types and will adapt favourably to those modalities or to individually prescribed training dosages. To our knowledge, there is one unpublished study that seeks to apply CCSs to this issue of exercise trainability, but no published research on the topic. Recent investigations have observed differential expression of microRNAs in high and low adapters to training [77,78]. Although detailed mechanistic studies on the interplay between mircroRNAs and chromosome conformation are limited, assessing changes in local chromatin organization around loci encoding microRNAs and the corresponding changes in microRNA expression could serve as an initial path for exploring this relationship.

### 7.3. CCSs and Nutrition

#### 7.3.1. Responses of CCS to Nutritional Stimuli

In addition to exercise itself, other exercise-related stimuli may be associated with CCSs. In sport and exercise, training interventions can be supported by nutritional strategies designed to promote adaptation, augment recovery and/or improve body composition [79]. Nutrition is an external stimulus prompting epigenetic modifications, in isolation and combined with exercise [80], leading us to suggest that CCSs may respond to dietary intervention and/or nutritional supplementation to regulate post-exercise gene expression. For example, the post-exercise increase in interleukin-6 (IL-6) gene expression is augmented under conditions of low muscle glycogen [81] and blunted by intra-exercise carbohydrate ingestion [82]. If CCSs rearrange in response to exercise stimuli to regulate gene expression, it is plausible that nutrition-induced changes in post-exercise gene expression are also associated with CCS responses. Investigating this hypothesis requires a single bout and/or intervention study with a second exercise condition, where a nutritional stimulus is imposed using dietary manipulation or supplementation. With stringent control of dietary intake and energy expenditure, such studies might offer insight into how CCSs respond to exercise-related nutritional stimuli.

#### 7.3.2. Using CCSs to Predict Responses or Adaptations to Nutritional Interventions

It is possible that pre-existing CCSs, as part of the established epigenetic landscape, are also involved in regulating responses and adaptations to nutritional interventions. Similar to the process of determining whether CCSs can prospectively regulate and predict adaptations to exercise training, observing individual responses to acute nutritional stimuli or adaptations to sustained nutritional interventions could predict how independent participants will respond under the same conditions. A similar example (not using CCSs) is how multiple factors such as blood parameters, dietary habits and physical activity were combined to predict glycaemic responses [83], while recent evidence in animal models suggests that chromatin interactions are, in turn, responsive to dietary stimuli [21]. To investigate the potential use of CCSs in similar settings, pre-intervention samples from participants exhibiting variable responses and adaptations to acute or chronic dietary changes could be screened to determine any associations with pre-intervention CCS. The ability of that CCS to predict how an independent population reacts to the same stimuli could then be evaluated.

### 7.4. CCSs and Environmental Extremes

Gene expression is affected by hot and cold climates [84,85,86] and conditions of low oxygen availability [87,88], though to our knowledge, the effects of such stimuli on CCSs are unknown. Heat acclimation is partially mediated by altered gene expression as an adaptation to peripheral haemodynamic load [89], and could be one example of an adaptation potentially associated with the reorganisation of specific CCSs. Initially, non-exercise studies are required to discover whether CCSs are regulated as part of such adaptations and, depending on their outcomes, may prompt further studies of whether CCSs are associated with the capacity to exercise in, and adapt to, extreme environments. If variable adaptations to the same environmental stimuli are observed between individuals, further analysis of pre-adaptation samples could determine if CCSs regulate and can predict the way an individual will respond to a particular environmental stimulus. Such information may be valuable to those training for sporting events held in extreme climates, or those with physically demanding occupations periodically based in extreme climates.

### 7.5. Use of CCSs to Detect Doping in Sport

The field of sports genomics has identified genes and polymorphisms associated with physical fitness phenotypes [7]. Whilst helping to understand the factors contributing to exercise performance, knowledge of key genomic loci may also encourage unscrupulous athletes to seek performance enhancement by delivering exogenous genetic material to the body, with the intention of increasing the production of proteins encoded by the genetic material [90]. At present, a key challenge in detecting such gene doping is differentiating artificial gene expression from that of normal physiological processes [91,92]. However, it is not known if specific patterns of expression, such as those associated with CCSs, relate to artificial gene expression, and whether such patterns could be used to differentiate between physiological and artificial profiles. The discovery of specific CCSs related to artificial gene expression could provide valuable blood-based screening tools in the pursuit of clean sport, though researching this area is ethically challenging.

### 7.6. Diagnostic Potential of CCSs Following Exercise-Related Trauma

Many sports are associated with injuries, with many internal and external factors influencing their occurrence [93]. Whilst many sports injuries and traumas are minor and resolve within weeks, others incur significant short- and long-term impairment. A noteworthy trauma is concussion, defined as a traumatically-induced transient disturbance of brain function that occurs when external forces of different intensities provoke rapid acceleration-deceleration of the brain [94]. Concussion occurs most frequently in contact sports and leads to a period of ‘metabolic brain vulnerability’ [95]. The duration of this period is highly variable from one concussion to another and can also vary between individuals [96], and secondary head trauma within this period seems particularly dangerous. Importantly, this period persists longer than measurable symptoms and often beyond the return of neuropsychological testing values to baseline, with many concussed patients being asymptomatic [97]. In light of these observations and the absence of structural damage from concussion, a number of approaches to measure the alterations in cellular function associated with concussion are proposed as diagnostic markers. Altered gene expression from sports-related concussion [98] leads us to suggest that a CCS might be associated with the regulation of genomic loci following concussive episodes and could discriminate between a concussed and non-concussed state. Indeed, the differentiation of physiological states by binary readout is a key characteristic of CCSs, meaning the presence or absence of a CCS has potential as a stable diagnostic marker for identifying molecular consequences of concussion. Such a marker may be particularly useful when symptoms have subsided or may never have been present at all. If combined with existing assessment protocols, detecting a molecular genomic signature of concussive injury would benefit physicians in the management and monitoring of athletes, allowing the temporary removal of affected athletes from competition and adding confidence when clearing unaffected athletes to return to action.

## 8. Conclusions

Initial applications of CCSs in biomedical research demonstrate promising findings, highlighting a potential new approach to investigating the epigenomic mechanisms underpinning exercise-related phenotypes. At present, we are not aware of published studies in the fields of sport and exercise science and medicine investigating how CCSs respond to external stimuli, or the regulatory role of CCSs in determining biological responses and adaptations. The features of CCSs described within the present review indicate the potential for this newly developed method to allow sport and exercise researchers to test a range of hypotheses related to personalised sport and exercise medicine. Appropriate design of exercise-related studies will allow researchers to address two very distinct applications of CCSs. Exploring the responsiveness of CCSs to acute and chronic stimuli can be investigated by comparing pre- and post-intervention states. In contrast, the regulatory role and prospect of a CCS to hold predictive ability can be investigated by identifying individuals who respond differently to an intervention and evaluating whether a specific pre-intervention CCS (i.e., *prospective* of the stimulus) is associated with the extent or nature of the subsequent response/adaptation. Thus, chromosome conformation presents an intriguing new potential direction for sport and exercise research.

## Figures and Tables

**Figure 1 genes-11-00905-f001:**
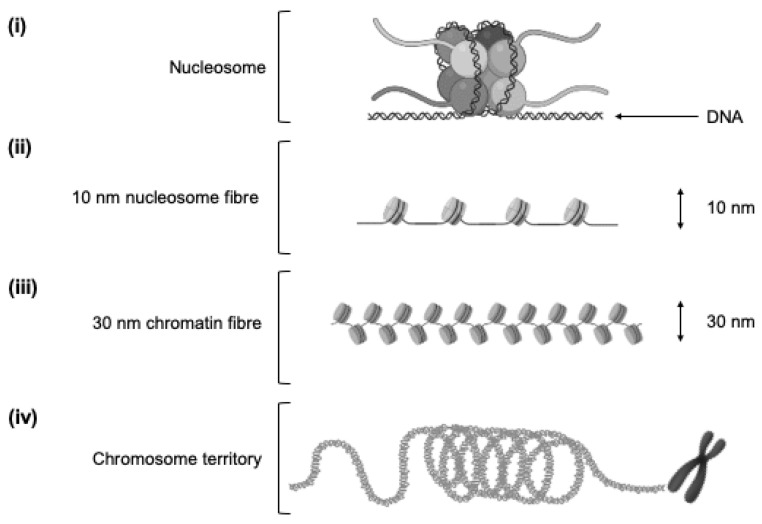
Nucleosomes and chromatin. (**i**) Nucleosome: Functional wrapping of 146 bp DNA strand around histone octamer (eight histone proteins). (**ii**) Ten nanometre nucleosome fibre: Repeating arrangement of individual nucleosomes linked together by continuous “linker” DNA sections approximately 20 bp long. (**iii**) Thirty nanometre chromatin fibre: Repeated folding of 10 nm fibre. (**iv**) Three hundred nanometre chromatin fibre: Repeated looping of 30 nm fibre to form loops averaging 300 nm in length. The 300 nm fibres are further compressed to produce a 700 nm × 250 nm fibre before being tightly coiled to produce the chromatid of a chromosome.

**Figure 2 genes-11-00905-f002:**
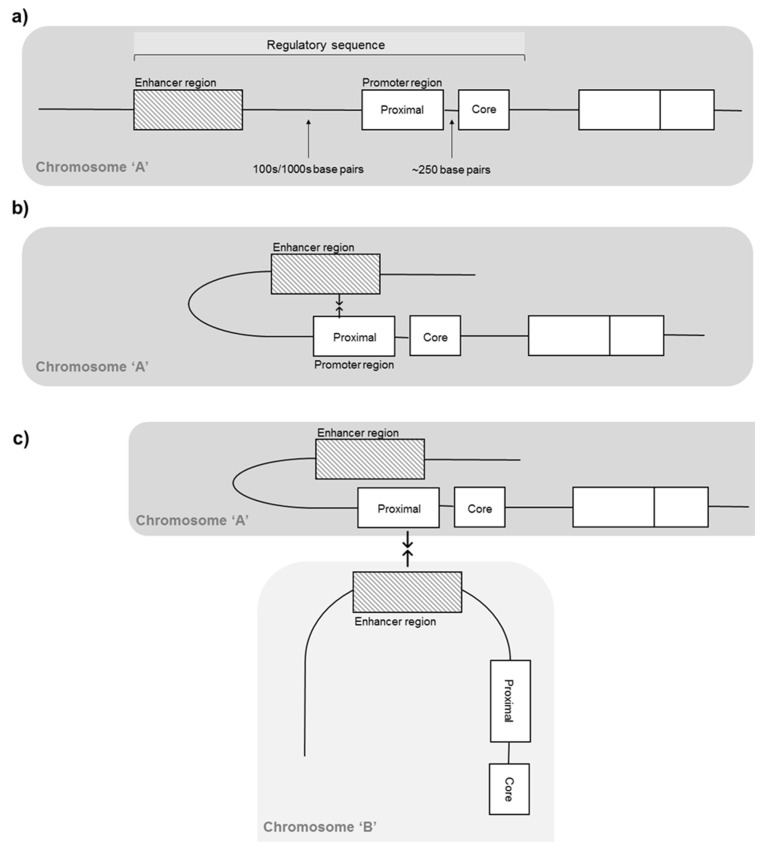
Functional genome organisation. Hypothetical example of three chromosomal arrangements from top to bottom. (**a**) Linear arrangement of an example gene, with non-adjacent enhancer and promoter regions separated by 100 s/1000 s of bp. (**b**) Intrachromosomal interaction. Looping of genomic sequence as a consequence of three-dimensional chromatin arrangement, bringing distal enhancer and promoter regions on the same chromosome into close physical proximity. (**c**) Interchromosomal interaction. Tight packaging of chromosomes within the nuclear space brings promoter and enhancer regions of different chromosomes into close physical proximity.

**Figure 3 genes-11-00905-f003:**
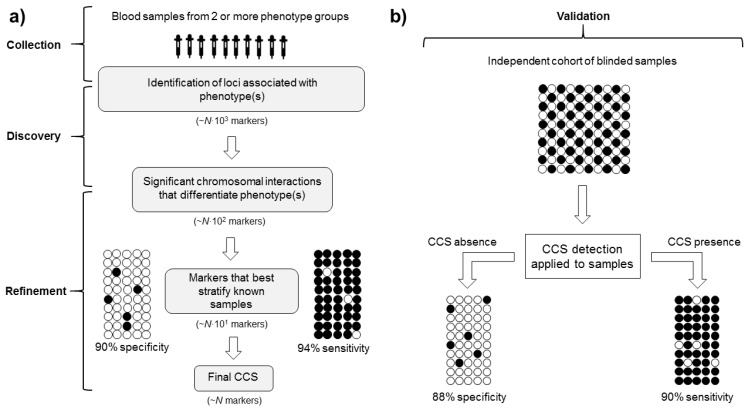
Defining a chromosome conformation signature (CCS) from sample collection to validation. (**a**) Discovery and refinement. Blood samples from phenotype groups (phenotype A = white dots, phenotype B = black dots) are screened for chromosomal interactions associated with each phenotype, with statistical refinement to determine the best discriminatory markers. Known samples are stratified to determine the sensitivity and specificity of the marker set. Specificity and sensitivity values are hypothetical. Typically, the number of markers in final CCS (*N*) would be ~5–15. (**b**) Validation. Blinded, independent samples are used to evaluate whether the final CCS identifies phenotype group correctly.

**Figure 4 genes-11-00905-f004:**
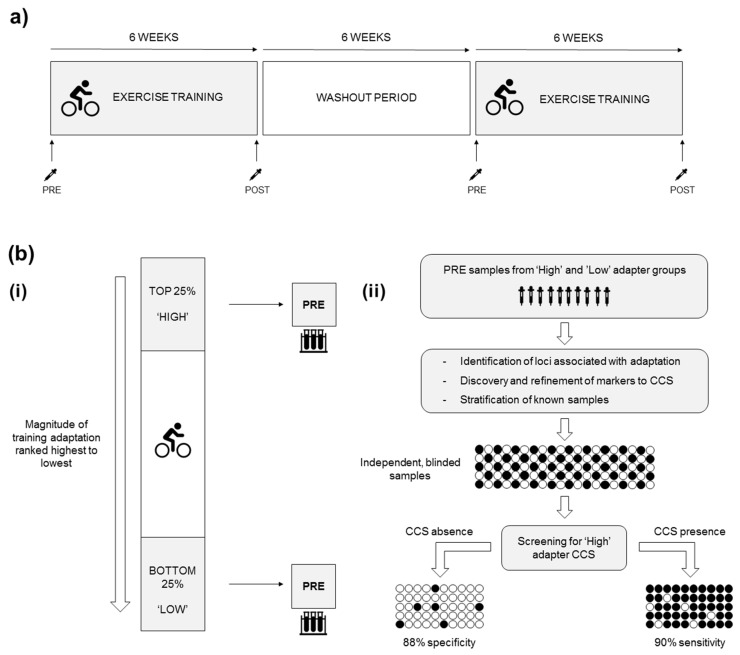
(**a**) Training intervention schematic. Study design of exercise training intervention to determine CCS associated with phenotypic adaptation to training. Peripheral blood collected before and after each 6-week intervention completed either side of a washout period. (**b**) Defining a CCS to predict training adaptation. (**i**) Determining adaptation differences. Participants ranked by magnitude of adaptation to the exercise intervention, with corresponding pre-training blood samples used for CCS discovery. (**ii**) CCS discovery using pre-training blood samples to discriminate between high and low adapters, with validation of CCS achieved by stratifying the predicted high and low adapters and subsequent observation of adaptations. Specificity and sensitivity values are hypothetical.

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
