# Peer review of "The Prospective Study of Epigenetic Regulatory Profiles in Sport and Exercise Monitored Through Chromosome Conformation Signatures"

_genes, 2020, doi:10.3390/genes11080905_

Round 1
Reviewer 1 Report
The authors submitted an interesting and well-written manuscript. In the first chapters, they introduce epigenetics and the measurement of chromosome conformation signatures (CCS) as new epigenetic biomarker. In the next chapters, the authors described the potential of CCS in different settings, for example for measuring the effect of acute exercise. The manuscripts introduces a new epigenetic aspect into the field of exercise science and is therefore highly recommended. Nevertheless, I have major concerns that should be addressed by the authors.
Major points:
- In the whole review, methodological aspects of CCS are missing. In a review, introducing a new measurement into a scientific field, a methodological discussion should be added. The new measurement should be discussed in the context of existing measurements (DNA-methylation, histone modification, miRNA) with pros and cons.
- The aim of the review is to discuss the potential of CSS in sport science. In my opinion, this aim is not reached yet. Especially in the later sections, the argumentation is always the same: differences in gene expression, CCS would be interesting. In addition, here should be a discussion with other epigenetic modifications. In chapter 4 the publication of Christova et al. Is cited, showing histone hyperacetylation after chromatin remodeling. This kinetic is of high relevance, but not considered in the rest of the review. A more detailed discussion of interplay with other epigenetic modifications should be added.
- In line 178, it is described that the change of gene expression after exercise may be due to changes in CCS. For other epigenetic modifications (like DNA-methylation or histone modifications) a lot of literature exists, that should be mentioned. Even histone modifications could represent a state of open chromatin structure and gene expression and should also be considered in CCS.
- Discussing CCS in the context of responders and non-responders should include literature on this topic so far. For example Ogasawara et al. 2016 investigated miRNA in the muscle of young men and found 26 miRNA differentially expressed between responders and non-responders. Would there be a benefit in measuring CCS rather than the 26 miRNA? This should be discussed.
- CCS should analyze the interaction of different genomic loci in the 3d structure. How precise the location of the loci could be described and could potentially DANN-methylation of interacting loci could also affect gene expression. During reading the review I was wondering, that CCS is more a new aspect of the epigenome that should be considered together with the other modifications.
Minor points:
- Figure 1: The numbering in the figure legend does not fit to the numbering in the figure.
Taking together, the review is very interesting and introduce a new epigenetic measurement into the field of exercise science. The argumentation should be based on existing studies and discussed in the context of other epigenetic modification (not just changes in gene expression). So far epigenetics and CCS are well described in the first sections of the manuscript, but the argumentation for CCS in exercise science is not sufficient so far. Since the topic is interesting and has potential to promote the inclusion of new methods to exercise science, I would recommend improving the manuscript.
Author Response
Please see the attachment/

Reviewer 2 Report
The present review by Hall et al focuses on the potential usefulness of the chromosome conformation capture (3C) technique and as a predictive biomarker of exercise adaptations. The idea of using 3C is that interaction frequencies between loci can be used to infer the spatial organization of the genome. This is to say that a specific genome conformation can be used to predict specific cellular outcomes. As such, the authors theorize that using this strategy, 3C signatures should be able to discriminate exercise responsive individuals and therefore serve as a predictive tool. While the topic is timely, we are in the "genomic era", the approach taken is somewhat biased. The review seems to focus on the EpiSwicth technology as the "go to" system to monitor genomic contacts, yet how this differs from standard 3C approaches is not discussed. Please describe the technology in detail to inform the reader about what constitutes this specific tool.
A major issue not discussed is the fact that genomic responses to exercise are dynamic in nature. So a static genome architectural feature will likely not be revealing of novel mechanisms that could be inferred otherwise using mRNA profiles.
Second, monitoring blood cells will likely not reflect the dynamic architecture of genomes of other tissues. Tissues are distinguished by gene expression patterns, and such distinct regulatory programs control tissue identity and adaptability. In fact, ~50% of all annotated genes show tissue specific expression (for reference see the GTEx Project). Using the same reasoning as the authors, once could say that if 3C profiles are similar between blood cells and tissues, the why not using mRNA from blood cells instead as a inference tool? This is a more mature and accessible technology than 3C.
Third, genome contacts represent only a small layer in gene expression and that do not account for neither the magnitude of gene expression, nor the splice isoforms of a particular gene product.
In summary, the review discusses a very timely topic that will inform the reader about 3C technologies and their use. The predictable value of this technology is however questionable as the data obtained will be associative in nature.
Round 2
Reviewer 1 Report
Dear authors,
The work is really improved by revisiting the articel.
Now the intention of the review is clear and I just have two minor points.
- Line 345: You suppose periferal blood as a good choice of biomaterial for investigations in exercise science. I think this is not completely true. For chronic training studies it is true but for acute studies the changes in blood cell homeostasis should be considered. Expacially in exercise science it is a problem in different studies (epigenetcs, immune function etc.) that changes in the the blood cell homeostasis are not considered in the interpretation of the results. It should be mentioned here.
- Line 377: Is this so clear that epigenetic marks are regulated by CCS? I think it is also possible that CCS are build up because of epigenetic marks. This should be phrased more carefully.
Author Response
Dear Reviewer,
Thank you for your constructive comments regarding our revised manuscript. We have made the changes that are detailed below:
Line 345: You suppose periferal blood as a good choice of biomaterial for investigations in exercise science. I think this is not completely true. For chronic training studies it is true but for acute studies the changes in blood cell homeostasis should be considered. Expacially in exercise science it is a problem in different studies (epigenetcs, immune function etc.) that changes in the the blood cell homeostasis are not considered in the interpretation of the results. It should be mentioned here.
Line 354-355: Text acknowledging that blood cell homeostasis must be considered and if possible, controlled for, particularly in acute exercise studies.
Line 377: Is this so clear that epigenetic marks are regulated by CCS? I think it is also possible that CCS are build up because of epigenetic marks. This should be phrased more carefully.
Line 376: Text making reference to the bi-directional association of other epigenetic marks with exercise responses, rather than that only being one directional.
We hope that these final changes are satisfactory and would like to thank you once again for your comments on our work. Your suggestions have improved our manuscript and we appreciate the time taken to review our work.
Dr Elliott Hall & co-authors.
Reviewer 2 Report
I am OK with the responses provided by the authors.Author Response
Dear Reviewer,
Thank you for your constructive feedback on our manuscript and for approving the changes we made following your comments. We appreciate the time taken to assist us in improving our work to meet the standard required for publication.
Kind regards,
Dr Elliott Hall & co-authors